Subject Category:
Biology (whole organism)

Subject Areas:
behaviour/biomechanics/fluid mechanics

Keywords:
vortex interactions, propulsion, bioengineering, jellyfish, plankton, biomechanics

Author for correspondence:

Brad J. Gemmell
e-mail: bgemmell@usf.edu

# A ctenophore (comb jelly) employs vortex rebound dynamics and outperforms other gelatinous swimmers

Brad J. Gemmell[1], Sean P. Colin[2,3], John H. Costello[2,4] and Kelly R. Sutherland[5]

[1]Department of Integrative Biology, University of South Florida, Tampa, FL 33620, USA
[2]Whitman Center, Marine Biological Laboratory, Woods Hole, MA 02543, USA
[3]Marine Biology/Environmental Sciences, Roger Williams University, Bristol, RI 02809, USA
[4]Biology Department, Providence College, Providence, RI 02908, USA
[5]Oregon Institute of Marine Biology, University of Oregon, Eugene, OR, USA

BJG, 0000-0001-9031-6591; KRS, 0000-0001-6832-6515

Gelatinous zooplankton exhibit a wide range of propulsive swimming modes. One of the most energetically efficient is the rowing behaviour exhibited by many species of schyphomedusae, which employ vortex interactions to achieve this result. Ctenophores (comb jellies) typically use a slow swimming, cilia-based mode of propulsion. However, species within the genus *Ocyropsis* have developed an additional propulsive strategy of rowing the lobes, which are normally used for feeding, in order to rapidly escape from predators. In this study, we used high-speed digital particle image velocimetry to examine the kinematics and fluid dynamics of this rarely studied propulsive mechanism. This mechanism allows *Ocyropsis* to achieve size-adjusted speeds that are nearly double those of other large gelatinous swimmers. The investigation of the fluid dynamic basis of this escape mode reveals novel vortex interactions that have not previously been described for other biological propulsion systems. The arrangement of vortices during escape swimming produces a similar configuration and impact as that of the well-studied 'vortex rebound' phenomenon which occurs when a vortex ring approaches a solid wall. These results extend our understanding of how animals use vortex–vortex interactions and provide important insights that can inform the bioinspired engineering of propulsion systems.

## 1. Introduction

Planktonic ctenophores typically use cilia, organized into ctene rows, for propulsion. However, members of the genus *Ocyropsis*

are known to rapidly escape disturbances by flapping their muscular oral lobes [1,2]. Although hydrodynamic interactions occurring during lobe flapping by *Ocyropsis* spp. have not been well studied, the similarities of gelatinous body form and broad body contractions shared with oblate scyphomedusae suggest analogous hydrodynamic patterns underlying propulsion by both groups. In this case, rowing propulsion by oblate scyphomedusae might serve as a useful model for understanding hydrodynamic processes powering escape swimming by *Ocyropsis* spp.

Rowing propulsion is used by oblate medusae to achieve one of the most energetically efficient means of swimming among animals (as low as $0.3 \, \mathrm{J \, kg^{-1} \, m^{-1}}$) [3,4]. Medusae do this without the benefit of powerful muscular arrays such as those found in other animal groups. In comparison to more advanced animal taxa, medusan muscles are poorly developed and capable of only very limited force production [5,6]. Instead, medusan swimming success relies upon highly coordinated production and manipulation of vortices along their bodies to generate pressure gradients that underlie the thrust forces enabling their energetically efficient swimming capabilities [7]. Both the contraction and recovery phases of rhythmic medusan swimming produce vortex arrangements that generate forward thrust for a medusa swimming along a linear pathway [4,8–10]. During bell contraction, fluid is pushed away from the bell, transferring momentum to the surrounding fluid and causing an oppositely directed push against the bell that moves it forward. Additionally, bending of the bell margin produces a vortex dipole on the outer side of the bell, creating a strong negative pressure region (suction zone) on the dorsal side of the bell simultaneously with the moderate positive pressure regions on the underside of the contracting bell to generate thrust for swimming [7,11]. These analytical results have demonstrated that this suction-dominant mechanism which occurs during contraction generates much of the propulsive thrust during swimming by the cosmopolitan medusa *Aurelia aurita*.

It is important to note that medusae do not rely solely upon the contraction phase to generate forward motion. Instead, they also employ vortex interactions during the recovery (i.e. relaxation) phase to advance their bodies through water [4,12]. As the bell returns to its original, pre-contraction state, the subumbrellar cavity is refilled with fluid that travels around the bell margin [13]. This fluid contains rotational energy and is known as the stopping vortex [14]. This vortex forces water against the inside surface of the bell, converting the rotational energy of the stopping vortices into forward body motion in a process termed passive energy recapture (PER). PER contributes as much as 60% of the net forward progress of the medusa during linear swimming [12]. The highly orchestrated production and alignment of vortices by the medusan bell enables both suction thrust and PER and, consequently, the highly efficient propulsion of this successful animal group.

Although energetically efficient, medusan rowing propulsion is generally not a rapid means of swimming. Instead, these species tend to be comparatively slow, cruising swimmers with limited escape abilities [9]. However, kinematic patterns of ctenophores within the genus *Ocyropsis* suggest that rowing swimming can be used for rapid propulsion. Similar to other lobate ctenophores, *Ocyropsis* spp. possess broad oral lobes and uses ciliary currents to swim. However, when startled, *Ocyropsis* flaps its broad lobes and rapidly escapes away from the disturbance [1] with a mean speed of $72 \, \mathrm{mm \, s^{-1}}$ for a distance of 1 m or more [2]. This mechanism has been observed to be successful in escaping attacks from a predatory ctenophore species in the genus *Beroe*. While unlikely to be successful in daylight against a visual predator, this escape behaviour may function against all predator types at night since *Ocyropsis* spp. are also known release a luminous mucus as part of the nighttime escape response which could act to confuse visual predators [2]. By contrast, the escape swimming in the ctenophore *Mnemiopsis leidyi*, which uses only ciliated ctene rows for propulsion, achieves less than half the speed of *Ocyropsis* [15].

While escape swimming in *Ocyropsis* appears to use rowing-type kinematics similar to that of medusae, the swimming performance of the ctenophore is much greater than that of rowing medusae. We quantified body kinematics combined with the fluid dynamics of this behaviour in order to determine how *Ocyropsis* can swim with such high proficiency. To accomplish this, we used high-speed, digital particle image velocimetry (DPIV) to quantify fluid interactions occurring during escape swimming of the oceanic ctenophore, *Ocyropsis maculata*. We present the first known case of an animal arranging vortices that resemble and function in the same manner as during the well-studied phenomena of vortex rebound. This process involves vortex generation and positioning such that opposite sign vortices interact to change the direction of the entire vortex superstructure surrounding the ctenophore. Vortex rebound has previously only been described from physical experiments and numerical simulations of interacting vortex rings; here, we discuss the potential implications for its use during animal swimming.

## 2. Methods

*Ocyropsis maculata* ctenophores were hand-collected in jars by SCUBA divers from waters off Santa Catalina, Panama (7°32′19.3″ N  81°29′11.1″ W) and immediately transported to the laboratory. All SCUBA plans were reviewed and permitted by the authors' institutional dive safety officers prior to commencing fieldwork. The animals were placed in glass filming vessels (30 × 10 × 25 cm) with field-collected water at *in situ* temperatures (26–28°C) within 6 h of collection for swimming and animal–fluid analyses. In order to elicit escape reactions, free-swimming ctenophores ($n = 5$) were gently touched on their aboral surface, which immediately caused the ctenophore to produce the escape swimming behaviour. This swimming behaviour was recorded using a high-speed digital video camera (SC1, Edgertronic) at 500 frames s$^{-1}$ at a resolution of 1280 × 1024 pixels. Only recordings of animals swimming upward were used in the analysis to eliminate the possibility of gravitational force aiding forward motion of the animal between pulses. Detailed two-dimensional kinematics were obtained using Image J v. 1.46 software (National Institutes of Health) to track the $x$ and $y$ coordinates of the apex (aboral end) and the moving lobe tips of the escaping ctenophore over time. Body swimming speeds and lobe tip speeds were calculated from the change in the position of the apex and lobe tip, respectively, over time, $t$, as

$$U = \frac{\left((x_2 - x_1)^2 + (y_2 - y_1)^2\right)^{1/2}}{t_2 - t_1}. \tag{2.1}$$

Using data from the literature on other gelatinous taxa, we compared peak swimming speeds to other large (greater than 2 cm) gelatinous swimmers. Body size was measured at the longest axis. Swimming data were obtained for another species of ctenophore (*M. leidyi*), two species of scyphomedusae (*Stomolophus meleagris* and *A. aurita*), two species of cubomedusae (*Chiropsella bronzie* and *Chironex fleckeri*) as well as a species of salp (*Cyclosalpa polae*). Swimming speeds were tested using one-way ANOVA to determine if a significant difference existed between means.

Alterations in body shape were quantified by the fineness ratio, $F$

$$F = \frac{h}{d}, \tag{2.2}$$

where $h$ is the bell height and $d$ is the bell diameter. The instantaneous fineness ratio, $F_i$, was measured at the midpoint of each interval used for measurement of ctenophore velocity. The fineness ratio of the body at its most laterally extended, uncontracted state corresponded closely to the minimum $F_i$ value, whereas full body contraction corresponded to the maximum $F_i$.

To quantify fluid interactions of escaping *O. maculata*, particle image velocimetry (PIV) analysis was performed whereby the ctenophores were illuminated with a laser sheet (532 nm, 1 W continuous wave) oriented perpendicular to the camera's optical axis to provide a distinctive body outline for image analysis and to ensure the animal remained in-plane, which ensures accuracy of two-dimensional estimates of position and velocity. The seawater containing the ctenophores was seeded with 10 μm hollow glass beads (LaVision Inc.). The velocities of particles illuminated in the laser sheet were determined from sequential images analysed using a cross-correlation algorithm (LaVision software). Image pairs were analysed with shifting overlapping interrogation windows of a decreasing size of 64 × 64 pixels to 32 × 32 pixels or 32 × 32 pixels to 16 × 16 pixels. To better understand the unique nature of this swimming mode, the fluid dynamics of swimming was quantified for another rowing gelatinous zooplankton, the moon jellyfish *A. aurita* ($n = 5$) as in [4,12]. The individuals examined were of similar size as *O. maculata* and ranged from 2.5 to 4 cm in diameter.

## 3. Results

### 3.1. Swimming performance

During escape behaviour, the oceanic ctenophore *O. maculata* achieved a maximum swimming speed of 125 mm s$^{-1}$ (s.d. 22, $n = 5$) (figure 1*b*). The peak accelerations during escape swimming were determined to be 720 mm s$^{-2}$ (s.d. 48, $n = 5$) (figure 1*c*). The maximum swimming speeds normalized by body size illustrate the extraordinary capabilities of this group of oceanic ctenophores when compared with other large (greater than 2 cm) gelatinous zooplankton. These swimming speeds (normalized by body length) are significantly higher (ANOVA, $p < 0.001$) than peak swimming speeds achieved in other large (greater

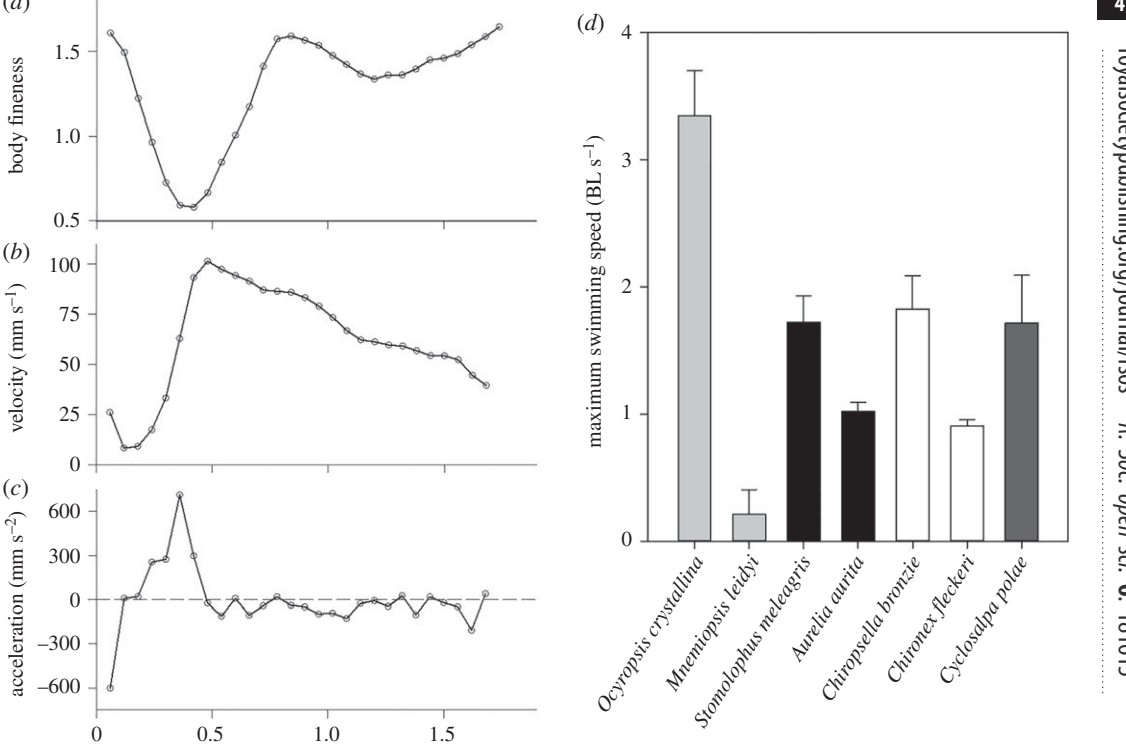

**Figure 1.** Swimming performance metrics. (*a*) Change in body fineness over time for a representative escape swimming sequence in the ctenophore *O. maculata*. (*b*) Instantaneous velocity over the escape sequence. (*c*) Instantaneous acceleration over the escape sequence. (*d*) Maximum attained swimming speeds in terms of body lengths per second for *O. maculata* and six other large (greater than 2 cm) gelatinous swimmers. Light grey bars, ctenophores; black bars, scyphomedusae; white bars, cubomedusae; dark grey bar, salp. Data for other gelatinous swimmers from: [3,4,15–17]. *Ocyropsis maculata* uses fast swimming only in short bursts, while other species listed tend to be continuous swimmers.

than 2 cm) gelatinous zooplankton swimmers (figure 1*d*). Compared to ciliary-based swimming that is typical of ctenophores, lobe swimming in *O. maculata* can achieve relative speeds more than an order of magnitude higher than the ctenophore *M. leidyi* (figure 1*d*). The gelatinous swimmer that comes closest to *O. maculata* in terms of relative swimming speed is the cubomedusa *C. bronzie*. This proficient jetting medusa reaches a mean peak speed of 1.82 BL s$^{-1}$ (s.d. 0.26). In comparison, *O. maculata* attains a mean peak speed of 3.35 BL s$^{-1}$ (s.d. 0.35) which is 84% higher than the fast swimming cubomedusa. It is interesting to note that of the most proficient swimmers from each group of large-bodied gelatinous zooplankton (scyphomedusae, cubomedusae and salps) all swim with peak relative speeds within 5% of each other and are not significantly different ($p = 0.434$). However, *O. maculata* is between 84 and 90% higher than these other swimmers. With a single flap of its lobes, *O. maculata* can also travel up to 120 mm. This swimming mode results in a drastically altered body profile during escape swimming. Fineness ratios range from 1.6 when at rest to 0.6 just prior to initiation of the contraction phase of the escape (figure 1*a*).

## 3.2. Lobe kinematics

In order to determine how *O. maculata* achieve their observed high accelerations and velocities, we quantified the kinematics of their lobes during escape reactions. Before an escape reaction is initiated, the lobes of *O. maculata* are oriented forward with the lobe tips in close proximity to each other (frame 1, figure 2*a*). When startled with a gentle touch, the ctenophores initiated their escape swimming cycle by rapidly expanding their lobes. This expansion can be characterized as flipping out the lobe tips quickly until the lobes reach the same plane (frame 3, figure 2*a*). At this point, the lobes begin immediately to contract. The contraction appears to be initiated at the base of the lobe which flares the lobe tips out laterally. The lobes then rapidly contract until the lobes are closer together than their initial position before the escape behaviour started.

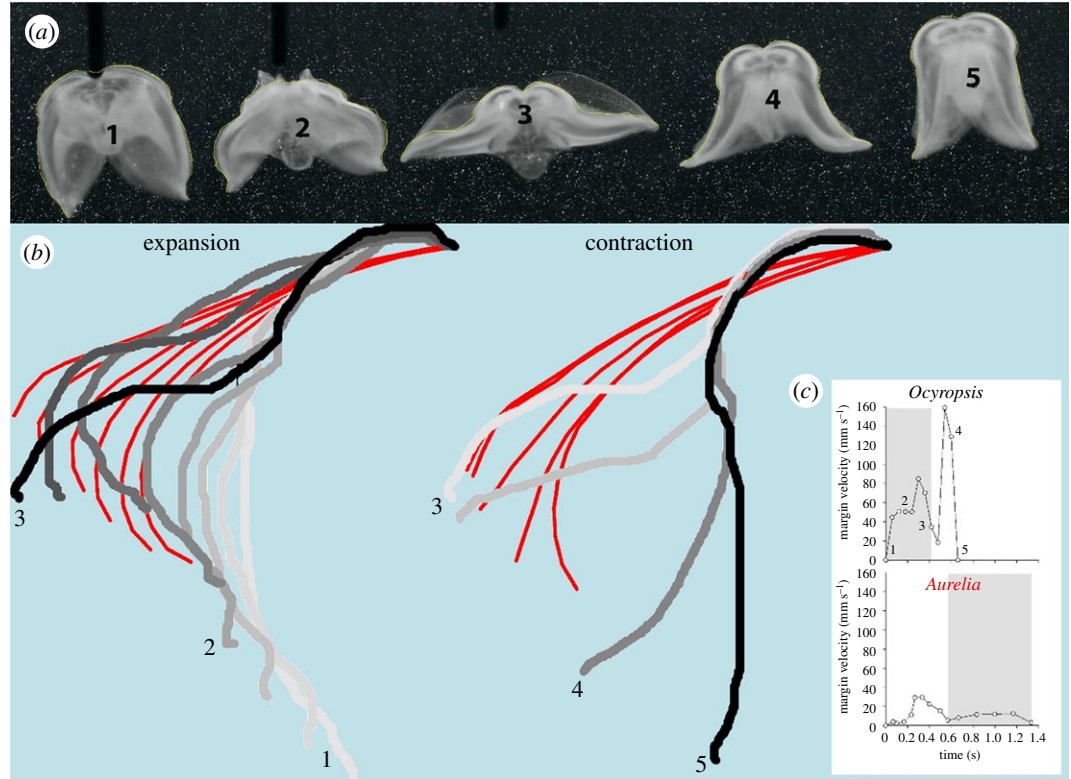

**Figure 2.** Lobe kinematics of *O. maculata* in comparison to the rowing medusa *A. aurita*. (*a*) Sequential images of *O. maculata* at different times during its swimming cycle. The cycle starts with lobe expansion (images 1–3) and finishes with a very rapid contraction (images 3–5). (*b*) An outline of the outer edge of the lobe shows the relative position of the *O. maculata* lobe (normalized to the apex) throughout the expansion and contraction phases. The bell outline (in red) of the scyphomedusae *A. aurita* underlays the *O. maculata* outline to illustrate the differences between *O. maculata* kinematics and that of a typical rowing medusae. The inset (*c*) shows the velocity of the tip of the lobe throughout the swim cycle. In essence, the lobe of *O. maculata* moves much faster (inset *c*) and over a much longer distance than typical rowing medusae.

To compare *O. maculata* kinematics to a typical rowing medusa, we overlaid the outlines of the lobe kinematics and the outline of the bell of the medusa *A. aurita* (red) throughout the swimming cycles (figure 2*b*). These outlines illustrate that the lobes of *O. maculata* move more extensively during the swim cycle than the bell margin of the rowing medusa (*O. maculata* moves 1.1 diameters versus *A. aurita* moves 0.43 diameters). Consequently, the lobe tip of *O. maculata* achieves much greater velocity during both the expansion and contraction phases than the bell tip of *A. aurita* (figure 2*c*) and completes the contraction phase of the swim cycle in approximately half the time of a comparably sized *A. aurita*. Additionally, since the lobes of *O. maculata* expand to a flat plane, there is no volume enclosed by the lobe at full expansion. By contrast, medusae do not expand as much and maintain a volume of fluid enclosed within the bell.

## 3.3. Flow fields

In order to be consistent with the identification of vortices produced by rowing organisms in the literature (e.g. [4,14]), we refer to the vortex that originates underneath the ctenophore during the expansion phase as the stopping vortex even though it forms at the beginning swim cycle. Likewise, the vortex that forms during the contraction phase of the rowing swimmers is referred to as the starting vortex.

To better understand the high performance during escape swimming in *O. maculata*, we quantified the instantaneous flow fields around the animals. Once stimulated to make an escape, *O. maculata* rapidly expanded its lobes, creating a stopping vortex underneath the lobes (figure 3*a,b*). Immediately prior to the rapid contraction phase of the lobes, the stopping vortex reached a peak vorticity of $22 \text{ s}^{-1}$ (s.d. 3). Initially, this vortex structure was located in a similar position to that of the stopping vortex created by the rowing schyphomedusa, *A. aurita*, during the relaxation phase of the swim cycle

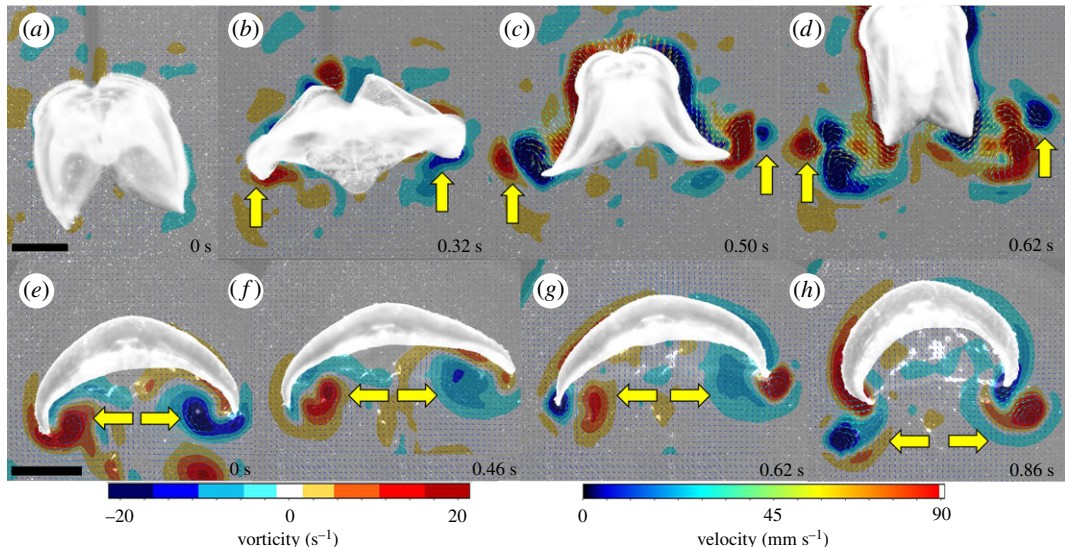

**Figure 3.** Fluid mechanics (velocity vectors and vorticity contours) from two gelatinous swimmers that employ rowing-based kinematics. (*a*–*d*) The ctenophore *O. maculata*. Note that the stopping vortex (indicated by yellow arrows) forms underneath the body during the expansion phase (*b*) but continues to move outside the lobes such that when the contraction begins, the stopping vortex is located outside of the newly formed starting vortex (*c*). The result of the vortex–vortex interaction is no net movement of the vortex superstructure (*d*). (*e*–*h*) The moon jellyfish *A. aurita*. Here, the stopping vortex (indicated by yellow arrows) remains confined under the jellyfish bell (*e*–*g*). The result is the downward movement of the entire vortex superstructure (*h*).

(figure 3*b,f*). However, as the lobe expansion of *O. maculata* continued, the stopping vortices moved laterally towards the lobe tips. This results in an important distinction during the contraction phase of the swim cycle between these two types of rowing swimmers.

As contraction of the lobes began, a starting vortex formed at the tips of both the ctenophore lobes and jellyfish bell margin (figure 3*c,g*). In the case of *A. aurita*, the stopping vortex interacts with the newly forming starting vortex underneath the animal and while the stopping vortex is positioned inside of the starting vortex (figure 3*g*). During the contraction phase of swimming in *O. maculata,* the stopping vortex also interacted with the newly forming starting vortex; however, in this case, the stopping vortex was located outside of the starting vortex due to greater lateral movement of the lobes during expansion (figure 3*c*). As the contraction phase of the swim cycle progressed through the point of vortex separation from the body, the strength of the stopping vortex did not subside in *O. maculata* (figure 3*d*).

In traditionally described rowing animals like *A. aurita* [5], the entire vortex superstructure is expelled in the wake and travels away from the animal (figure 3*h*). However, the interaction of opposite sign vortices in the arrangement produced by *O. maculata* appears to prevent the backwards movement of the entire vortex superstructure and stretches the starting vortex until it pinches off into two distinct vortices (figure 3*d*). Once separated from downward movement of the lobes, the stopping vortex, still located outside of the starting vortex, demonstrated its ability to exert an upwards pull on the entire vortex superstructure (figure 4).

## 4. Discussion

In terms of energetic efficiency and the cost of transport of locomotion, gelatinous zooplankton are some of the best performers on the planet [3,4,16]. However, in terms of swimming proficiency, gelatinous zooplankton are often overlooked as they are outperformed by other groups such as fish and squid [18–21]. Yet, there are some taxa of gelatinous zooplankton which, relative to body size, come close to the peak speeds reached by other groups [22,23]. Among the larger (greater than 2 cm) gelatinous species, top performers in terms of swimming speed are members of the rhizostomae, cubozoa and the salps. Ctenophores, using a cilia-based propulsion system, swim an order of magnitude slower than these more proficient groups (figure 1). These cilia are fused into rows of ctene plates with plates consisting of thousands of individual cilia. The ctenes then beat metachronally to move the animal

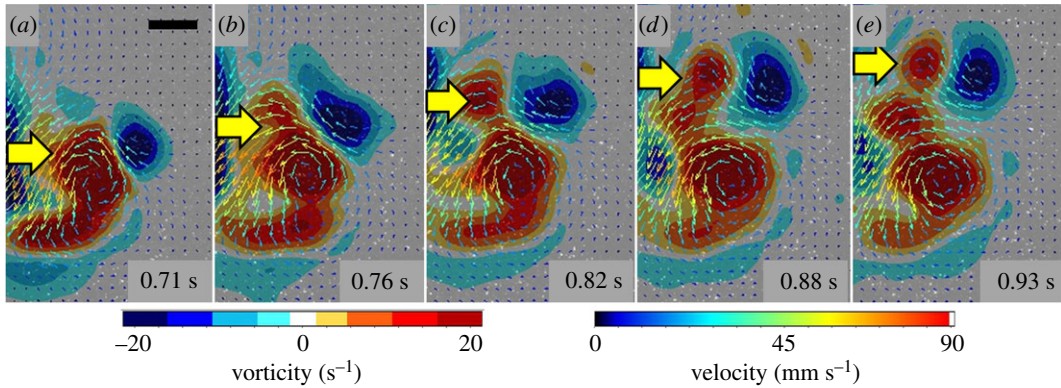

**Figure 4.** Fluid mechanics (velocity vectors and vorticity contours) of one side of the vortex superstructure shed by *O. maculata*. Note that while the animal moved its lobes and the resulting starting vortex in a downward direction, the starting vortex (red) is simultaneously pulled upwards (as indicated by yellow arrows) due to interaction with the opposite sign stopping vortex (blue). The result is a vertically elongated vortex superstructure.

forwards. Our investigation into the lobe-based swimming displayed by the oceanic ctenophore *O. maculata* finds that this species can achieve relative speeds and accelerations which are significantly greater than other proficient gelatinous swimmers (figure 1).

This high level of performance is achieved by using rowing-based kinematics commonly observed in schyphomeusae, but with some important differences. One of the most apparent of these differences is the change in body fineness over the course of a swim cycle. *Ocyropsis maculata* has a fineness ratio that ranges from 1.6 to 0.6 (figure 1), whereas *A. aurita* swims with a fineness ratio that varies by only 0.2, from 0.5 to 0.3 [24]. The larger change in ctenophore body conformation over the course of the swim cycle reflects a greater range of movement of the lobes relative to a medusa bell (figure 2). Unlike cnidarian medusae, ctenophore muscle is not constrained to a single layer [25]. This may allow for the greater range of motion that would accelerate more water and lead to greater thrust production [26,27]. The extended range of ctenophore motion relative to other rowing gelatinous swimmers has additional implications for how the resulting vortices are positioned and interact with one another.

Since the lobes of *O. maculata* expand beyond that of medusae to a flat plane (figure 2), there is no volume enclosed by the lobes at full expansion. This allows the stopping vortex formed during the expansion phase to extend beyond the lobes prior to the initiation of the contraction phase. One consequence of the *O. maculata* stopping vortex not remaining underneath the body is that there can be no benefit of PER as seen in medusae. PER provides additional thrust without the need for additional energetic expenditure of body movements [4]. In the case of *Ocyropsis*, this trade-off can be considered in the light of the different ecological roles of swimming between the two groups of animals. Medusae swim continuously and thus there will be strong selective pressure to employ tactics that minimize energetic expenditures and cost of transport. In this case, medusae can use the slow developing PER mechanism in which an extended pause prior to contraction provides a substantial benefit in terms of cost of transport [12]. By contrast, *Ocyropsis* spp. uses a rowing mode of swimming for escape and possibly to re-position themselves periodically [2,28]. Here, a substantial pause prior to the contraction phase would defeat the purpose of a rapid escape and so the ctenophore uses the rotational energy stored in the stopping vortex a different way.

By positioning the stopping vortex outside of the lobes, the contraction phase generates a new vortex of opposite sign (starting vortex) just inside of the existing stopping vortex (figure 3c). This leads to strong interaction between the pair of vortices in a configuration that closely resembles the vortex arrangement during a phenomenon known as 'vortex rebound' [29]. Vortex rebound is known from investigations of a vortex ring approaching a solid, flat surface at a direction normal to the axis of the ring. At a particular distance from the wall, the axial velocity changes direction and the ring moves rapidly away from the surface [30–34]. This reversal of the axial velocity is commonly referred to as the vortex rebound. It has been experimentally determined that the rebound effect is due to a secondary vortex that is produced at the solid boundary and subsequently interacts with the original vortex to rapidly lift the entire vortex superstructure upwards [35].

The distinction between traditionally described vortex rebound in the literature and that observed with the ctenophore appears to be simply how and when the two interacting, opposite sign vortices are

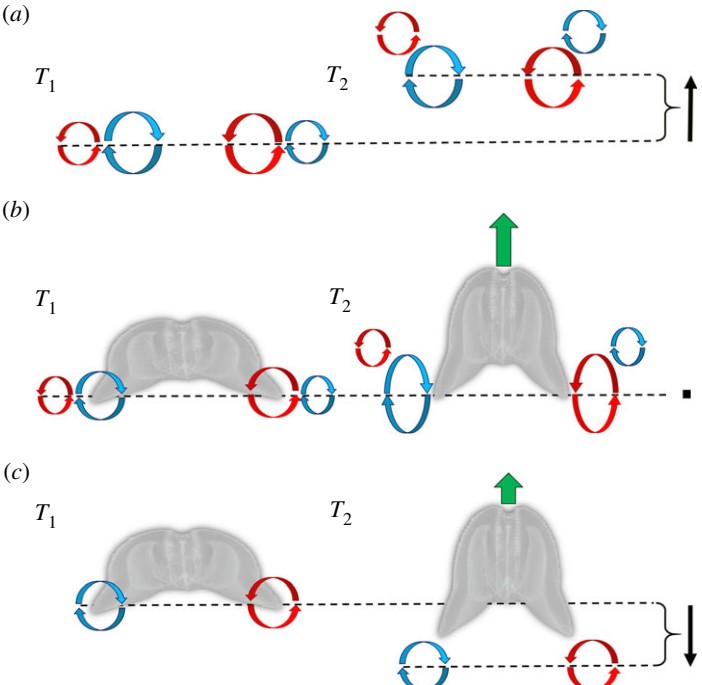

**Figure 5.** Conceptual figure illustrating a proposed mechanism for enhancing biological propulsion through the arrangement of vortices into a 'vortex rebound' configuration. (*a*) Vortex–vortex interactions cause a rebound effect, which pulls the entire vortex superstructure upwards (difference between the dotted lines). Based on data from Orlandi [29]. (*b*) Vortex configuration observed during escape swimming in the ctenophore *O. maculata*. Note that instead of the vortex superstructure being pulled upwards, the starting (inner) vortex is stretched as the animal pushes downwards with its lobes. This could result in greater overall thrust for the animal. (*c*) A hypothetical case based on medusae rowing-type swimming where the starting vortex is ejected backwards into the wake. Here, the reaction force on the lobes would be lower than that in (*b*) and would lead to lower thrust. Black arrows show net motion of vortex superstructure; green arrows show net thrust of ctenophore.

produced. In the case of traditionally described vortex rebound, the secondary vortex ring is formed later and ejected from the boundary layer near the solid wall [34]. In the case of the ctenophore, the secondary vortex forms first, and is ejected during the expansion phase of the escape. Next, what is traditionally described as the primary vortex is created by the contraction of the ctenophore lobes (figure 3). The end result appears to be the same with the outer vortex interacting and exerting an upwards force on the entire vortex superstructure which advects the structure forward/upward (figure 4).

How might a vortex rebound configuration aid in ctenophore escape swimming? To understand the potential adaptive significance of such a vortex arrangement, it may be useful to consider the differences in jumping performance on solid versus a deformable surface. Even with compensatory kinematics, jump performance on sand is significantly lower than jumps that occur on a rigid surface [36,37]. In general, the more deformable the substrate surrounding the propulsor, the lower the thrust that can be generated due to a reduced reaction force. Water is a highly deformable substrate and thus presents challenges for aquatic swimmers in order to generate a sufficient reaction force. The generalized explanation for swimmers has been that the propulsive element (a lobe in the case of a ctenophore) moves backward and generates a force that increases the momentum of the water passing backward [26]. An equal opposing force (the reaction force) is subsequently exerted by the water on the propulsive element to generate thrust and move the animal forward. But what if the water that was being pushed backwards could resist some of this motion? It would generate a much higher reaction force. Since the vortex rebound interaction produced by the ctenophore advects the vortex superstructure forward (figure 4), it seems likely that the fluid would produce a higher reaction force as the propulsor moves backwards within a vortex superstructure (figure 5).

## 5. Conclusion

The ability of a lobate ctenophore to reach speeds that can exceed those of other large gelatinous swimmers relies on alterations to the previously described 'rowing' kinematics displayed by many

schyphozoan and some hydrozoan medusa [8,14]. These kinematic alterations allow for a different arrangement of vortices that create a 'vortex rebound' effect. This fluid phenomena is well documented in the physics literature [30–35], but to our knowledge, this type of vortex arrangement and the resulting phenomena of the upwards movement of the vortex superstructure have not previously been documented in biological propulsion. By arranging fluid in this manner both biological systems and engineered underwater vehicles may benefit from an increased reaction force that may significantly enhance the ability to accelerate a body under water. Further investigation into how the vortex rebound effect may contribute to overall thrust is needed, but these findings further our understanding into how animals can use and take advantage of vortex–vortex interactions and may provide important insights that can inform the bioinspired engineering of propulsion systems.

Data accessibility. The raw data that were presented and analysed for this manuscript are available at: https://figshare.com/s/2746dabe807aa03ed274.

Authors' contributions. All authors participated in the design of the study. B.J.G. and K.R.S. collected the PIV data. B.J.G., S.P.C. and J.H.C. analysed the data. B.J.G. wrote the first draft and all authors contributed to revisions. All authors gave final approval for publication.

Competing interests. We declare we have no competing interests.

Funding. This research was supported by the grants from the National Science Foundation UNS-1511996 and IDBR-1455471 to B.J.G., S.P.C. and J.H.C. as well as OCE-1829945 to B.J.G., S.P.C., J.H.C. and K.R.S.

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
