## [Reviewer comments · Royal Society Open Science]

Review History

RSOS-181615.R0 (Original submission)

Review form: Reviewer 1

Is the manuscript scientifically sound in its present form?

Yes

Are the interpretations and conclusions justified by the results?

Yes

Is the language acceptable?

Yes

Is it clear how to access all supporting data?

Yes

Do you have any ethical concerns with this paper?

No

Have you any concerns about statistical analyses in this paper?

No

Recommendation?

Accept with minor revision (please list in comments)

Comments to the Author(s)

Please find attached my review of the paper "A ctenophore (comb jelly) employs vortex rebound dynamics and outperforms other gelatinous swimmers" for consideration for publication in Royal Society Open Science. In this paper, the authors use particle image velocimetry to quantify the propulsive flows generated by the ctenophore *Ocyropsis*. In particular, the authors consider swimming mode in which the lobes of the comb jelly are "rowed" (in addition to any flows generated by the ciliary plates). They find that during the escape mode, this comb jellyfish can reach dimensionless speeds nearly double that of other gelatinous animals. The fluid dynamic mechanism behind the swimming behavior appears to be representative of a vortex rebound mechanism. Comparisons are made to the jellyfish *A. aurita* which also uses a rowing or paddling mode of propulsion that seems similar to the comb jellyfish escape swimming mode. Interesting differences in contraction kinematics and the resulting wake can be seen, however, which explain the difference in swimming speeds. The experimental work in the paper is of solid quality, the results are clearly presented, and the work should be of broad interest to the biomechanics and biofluids communities.

Minor comments

- 1) It is stated in the abstract and the discussion that the results might inform the bio-inspired design of propulsive systems. Some elaboration on this point seems worthwhile.
- 2) Figure 3 and Supplemental Figure 1 use a very similar color map for the vorticity and for the velocity vectors. It would be preferable to use some other color scheme. It is very difficult to see the velocity vectors in Figure 3.
- 3) Additional discussion of the flows generated by the ciliary plates would help contrast the escape mode from other propulsive modes in this species and other comb jellyfish. It is likely worth explaining further that these are ciliary fused plates rather than individually spinning cilia.

Review form: Reviewer 2**Is the manuscript scientifically sound in its present form?**

Yes

Are the interpretations and conclusions justified by the results?

Yes

Is the language acceptable?

Yes

Is it clear how to access all supporting data?

Yes

Do you have any ethical concerns with this paper?

No

Have you any concerns about statistical analyses in this paper?

No

Recommendation?

Accept with minor revision (please list in comments)

Comments to the Author(s)

Major;

None

Minor;

-In the introduction the authors consistently talk about the highly efficient of swimming in certain medusa, I would like to see some values included to give this claim of efficiency some context.

-Pg. 7, Line 15 appears to be missing 'of the'.

-Pg. 7, Line 38; 'newly forming starting vortex as while the stopping vortex remained', sounds odd, consider revising.

-Pg.7, Line 56; 'in' should be 'it'.

- How did the authors measure body length for the individual species? What size were the animals that the proficiency measurements were being compared to?

-The comparison to other gelatinous swimmers is valid and interesting, however, I think that more context needs to be given to the different types of propulsion system being looked at. i.e, Ocyropsis may well have a maximum normalised body speed of twice that of Stomolophus meleagris, but ecologically speaking they are completely different locomotor modes, with one being an escape response and one being a continuous cruising style of swimming. I understand the authors are looking at proficiency, but I feel as the manuscript is written now it is hard to know which species are cruisers and which are an escape swim. Something as simple as an extra line in the figure legend would help clarify this.

- In the discussion the authors talk about the increased range of motion in *O. maculate* being bought about by increased muscle fibre layers. Have the authors considered how changes in the material properties of the mesoglea might affect the range of motion achievable in these species?

-How does the musculature compare between Ocyropsis and other ctenophores that do not perform these escape swims?

-In the videos of the swims the animals often appear very close the wall of the aquarium, are the authors confident this did not affect the measurements?

-As a reader I would also be keen to see a video showing multiple swimming strokes at normal speed.

Decision letter (RSOS-181615.R0)

04-Jan-2019

Dear Dr Gemmell

On behalf of the Editors, I am pleased to inform you that your Manuscript RSOS-181615 entitled "A ctenophore (comb jelly) employs vortex rebound dynamics and outperforms other gelatinous swimmers" has been accepted for publication in Royal Society Open Science subject to minor revision in accordance with the referee suggestions. Please find the referees' comments at the end of this email.

The reviewers and handling editors have recommended publication, but also suggest some minor revisions to your manuscript. Therefore, I invite you to respond to the comments and revise your manuscript.

- Ethics statement

- Data accessibility

If you wish to submit your supporting data or code to Dryad (<http://datadryad.org/>), or modify your current submission to dryad, please use the following link:
<http://datadryad.org/submit?journalID=RSOS&manu=RSOS-181615>

- Competing interests

- Authors' contributions

- Acknowledgements

- Funding statement

Because the schedule for publication is very tight, it is a condition of publication that you submit the revised version of your manuscript before 13-Jan-2019. Please note that the revision deadline will expire at 00.00am on this date. If you do not think you will be able to meet this date please let me know immediately.

- 1) A text file of the manuscript (tex, txt, rtf, docx or doc), references, tables (including captions) and figure captions. Do not upload a PDF as your "Main Document";
- 2) A separate electronic file of each figure (EPS or print-quality PDF preferred (either format should be produced directly from original creation package), or original software format);
- 3) Included a 100 word media summary of your paper when requested at submission. Please ensure you have entered correct contact details (email, institution and telephone) in your user account;
- 4) Included the raw data to support the claims made in your paper. You can either include your data as electronic supplementary material or upload to a repository and include the relevant doi within your manuscript. Make sure it is clear in your data accessibility statement how the data can be accessed;

5) All supplementary materials accompanying an accepted article will be treated as in their final form. Note that the Royal Society will neither edit nor typeset supplementary material and it will be hosted as provided. Please ensure that the supplementary material includes the paper details where possible (authors, article title, journal name).

on behalf of Professor Brooke Flammang (Associate Editor) and Kevin Padian (Subject Editor)
openscience@royalsociety.org

Associate Editor Comments to Author (Professor Brooke Flammang):

Both reviewers find this manuscript to be a significant contribution and suggest minor edits for clarification. Please incorporate their suggestions into your revision.

Reviewer comments to Author:
Reviewer: 1

Comments to the Author(s)

Please find attached my review of the paper "A ctenophore (comb jelly) employs vortex rebound dynamics and outperforms other gelatinous swimmers" for consideration for publication in Royal Society Open Science. In this paper, the authors use particle image velocimetry to quantify the propulsive flows generated by the ctenophore *Ocyropsis*. In particular, the authors consider swimming mode in which the lobes of the comb jelly are "rowed" (in addition to any flows

generated by the ciliary plates). They find that during the escape mode, this comb jellyfish can reach dimensionless speeds nearly double that of other gelatinous animals. The fluid dynamic mechanism behind the swimming behavior appears to be representative of a vortex rebound mechanism. Comparisons are made to the jellyfish *A. aurita* which also uses a rowing or paddling mode of propulsion that seems similar to the comb jellyfish escape swimming mode. Interesting differences in contraction kinematics and the resulting wake can be seen, however, which explain the difference in swimming speeds. The experimental work in the paper is of solid quality, the results are clearly presented, and the work should be of broad interest to the biomechanics and biofluids communities.

Minor comments

- 1) It is stated in the abstract and the discussion that the results might inform the bio-inspired design of propulsive systems. Some elaboration on this point seems worthwhile.
- 2) Figure 3 and Supplemental Figure 1 use a very similar color map for the vorticity and for the velocity vectors. It would be preferable to use some other color scheme. It is very difficult to see the velocity vectors in Figure 3.
- 3) Additional discussion of the flows generated by the ciliary plates would help contrast the escape mode from other propulsive modes in this species and other comb jellyfish. It is likely worth explaining further that these are ciliary fused plates rather than individually spinning cilia.

Reviewer: 2

Comments to the Author(s)

Major;

None

Minor;

-In the introduction the authors consistently talk about the highly efficient of swimming in certain medusa, I would like to see some values included to give this claim of efficiency some context.

-Pg. 7, Line 15 appears to be missing 'of the'.

-Pg. 7, Line 38; 'newly forming starting vortex as while the stopping vortex remained', sounds odd, consider revising.

-Pg.7, Line 56; 'in' should be 'it'.

- How did the authors measure body length for the individual species? What size were the animals that the proficiency measurements were being compared to?

-The comparison to other gelatinous swimmers is valid and interesting, however, I think that more context needs to be given to the different types of propulsion system being looked at. i.e, *Ocyropsis* may well have a maximum normalised body speed of twice that of *Stomolophus meleagris*, but ecologically speaking they are completely different locomotor modes, with one being an escape response and one being a continuous cruising style of swimming. I understand the authors are looking at proficiency, but I feel as the manuscript is written now it is hard to know which species are cruisers and which are an escape swim. Something as simple as an extra line in the figure legend would help clarify this.

- In the discussion the authors talk about the increased range of motion in *O. maculate* being

bought about by increased muscle fibre layers. Have the authors considered how changes in the material properties of the mesoglea might affect the range of motion achievable in these species?

-How does the musculature compare between Ocyropsis and other ctenophores that do not perform these escape swims?

-In the videos of the swims the animals often appear very close the wall of the aquarium, are the authors confident this did not affect the measurements?

-As a reader I would also be keen to see a video showing multiple swimming strokes at normal speed.

Author's Response to Decision Letter for (RSOS-181615.R0)

See Appendix A.

Decision letter (RSOS-181615.R1)

31-Jan-2019

Dear Dr Gemmell,

I am pleased to inform you that your manuscript entitled "A ctenophore (comb jelly) employs vortex rebound dynamics and outperforms other gelatinous swimmers" is now accepted for publication in Royal Society Open Science.

on behalf of Professor Brooke Flammang (Associate Editor) and Professor Kevin Padian (Subject Editor)

Follow Royal Society Publishing on Twitter: [@RSocPublishing](https://twitter.com/RSocPublishing)

Appendix A

Response to reviewer comments:

Authors' overview: *We thank the reviewers for their time and helpful comments. We have made the suggested changes to improve the clarity to readers and feel that the manuscript has been significantly improved as a result. The specific changes made are outlined below.*

Reviewer: 1

Comments to the Author(s)

Minor comments

1) It is stated in the abstract and the discussion that the results might inform the bio-inspired design of propulsive systems. Some elaboration on this point seems worthwhile.

Authors: *We have added some text to the conclusion that addresses how bio-inspired underwater vehicles may benefit from employing the fluid/vortex arrangement that has been identified in this study.*

2) Figure 3 and Supplemental Figure 1 use a very similar color map for the vorticity and for the velocity vectors. It would be preferable to use some other color scheme. It is very difficult to see the velocity vectors in Figure 3.

Authors: *We have gone back to the figure data and played around with several other color schemes in Lavis's DaVis software program and they all give similar looking results. We also tried to make the vectors thicker and more pronounced as well as making all the vectors a unicolor black shade. This resulted in the obscuring of the vorticity color palette (which is the main focus of the figure). Ultimately, we feel the problem of clearly seeing the vector field was mostly likely the result of the low resolution of the figure in the initial submission. We have replaced all figures in the revision with high resolution copies and feel that this resolves the issue with not being able to clearly resolve the velocity vectors.*

3) Additional discussion of the flows generated by the ciliary plates would help contrast the escape mode from other propulsive modes in this species and other comb jellyfish. It is likely worth explaining further that these are ciliary fused plates rather than individually spinning cilia.

Authors: *we have added text in the first paragraph of the discussion that reflects the reviewer's comment about clarity of the cilia-based mode locomotion found in all ctenophores, including Ocyropsis.*

Reviewer: 2

Comments to the Author(s)

Minor;

-In the introduction the authors consistently talk about the highly efficient of swimming in certain medusa, I would like to see some values included to give this claim of efficiency some context.

Authors: *We have added that jellyfish can achieve energetic swimming efficiencies as low as $0.3 J kg^{-1} m^{-1}$ to the second paragraph of the introduction.*

-Pg. 7, Line 15 appears to be missing 'of the'.

Authors: *Fixed.*

-Pg. 7, Line 38; 'newly forming starting vortex as while the stopping vortex remained', sounds odd, consider revising.

Authors: *This sentence has been reworded.*

-Pg.7, Line 56; 'in' should be 'it'.

Authors: *Fixed.*

- How did the authors measure body length for the individual species? What size were the animals that the proficiency measurements were being compared to?

Authors: *We have added details to the methods to clarify the point that body size was measured at the longest axis and length data on all species is available in the online data depository.*

-The comparison to other gelatinous swimmers is valid and interesting, however, I think that more context needs to be given to the different types of propulsion system being looked at. i.e, Ocyropsis may well have a maximum normalised body speed of twice that of Stomolophus meleagris, but ecologically speaking they are completely different locomotor modes, with one being an escape response and one being a continuous cruising style of swimming. I understand the authors are looking at proficiency, but I feel as the manuscript is written now it is hard to know which species are cruisers and which are an escape swim. Something as simple as an extra line in the figure legend would help clarify this.

Authors: *We appreciate this point and as suggested have added a line to the figure legend for clarity.*

- In the discussion the authors talk about the increased range of motion in *O. maculate* being brought about by increased muscle fibre layers. Have the authors considered how changes in the material properties of the mesoglea might affect the range of motion achievable in these species?

Authors: *The reviewer brings up an interesting point. While we are unable to address the question in detail, we added text that speculates there could be material properties other than muscle that allow for the greater range of motion.*

-How does the musculature compare between *Ocyropsis* and other ctenophores that do not perform these escape swims?

Authors: *We are unaware of any detailed physiological descriptions of oceanic ctenophores, so we are unable to provide any insight into this interesting question.*

-In the videos of the swims the animals often appear very close to the wall of the aquarium, are the authors confident this did not affect the measurements?

Authors: *While some swimming sequences occurred with a small part of the ctenophore body somewhat close to a wall, we are very confident that this did not affect any of our measurements for the following reasons: 1) We did not use data from any animals touching a wall. 2) The swimming performance for the few individuals that were close to the wall was not significantly different than those in the middle of the filming tank. 3) Fluid structures (vortices) that formed somewhat close to a wall, did not appear altered or to dissipate any faster than those in the middle of the tank.*

-As a reader I would also be keen to see a video showing multiple swimming strokes at normal speed.

Authors: *Unfortunately in all of our underwater field videos (the only ones with a field of view large enough to capture multiple swimming strokes) the animals quickly swam out of plane/focus and so we do not have any high quality videos of this.*